# UNDERSTANDING AND ADDRESSING SPURIOUS CORRELATION VIA NEURAL TANGENT KERNELS: A SPECTRAL BIAS PERSPECTIVE

## ABSTRACT

The existence of spurious correlations can prompt neural networks to depend heavily on features that exhibit strong correlations with the target labels exclusively in the training set, while such correlations may not persist in real-world scenarios. As a consequence, this results in suboptimal performance within certain subgrouping of the data. In this work, we leverage the theoretical insights of the Neural Tangent Kernel (NTK) to investigate the group robustness problem in the presence of spurious correlations. Specifically, we identify that poor generalization is not solely a consequence of statistical biases inherent in the dataset; rather, it also arises from the disparity in complexity between spurious and core features. Building upon this observation, we propose a method that adjusts the spectral properties of neural networks to mitigate bias without requiring knowledge of the spurious attributes.

## 1 INTRODUCTION

Deep neural networks (DNNs) have become exceptionally powerful tools for various tasks, ranging from image recognition to natural language processing. Their ability to learn intricate patterns and extract high-level representations from complex data has revolutionized the field of machine learning. However, despite their impressive capabilities, DNNs also pose challenges in several domains. One such challenge is the presence of spurious correlation within DNNs. Spurious correlations refer to the scenario where certain (potentially simpler) task-irrelevant attributes in the training set are highly correlated with the target labels. For example, consider the scenario where a DNN is trained to distinguish between images containing cars and bicycles. In the training dataset, an unintended sampling bias might emerge, leading to a situation where the majority of car images happen to be predominantly of a particular color, say blue, while the majority of bicycle images tend to have a different color, like red. This sampling bias inadvertently introduces a spurious correlation between the object category and the color attribute. Consequently, the trained DNN may mistakenly learn to associate the presence of a certain color with a particular object class, leading to erroneous predictions when faced with images featuring cars or bicycles of different colours.

Spurious correlations can have significant implications in real-world applications. Relying on these false associations can result in flawed predictions, inaccurate analyses, and misguided actions, particularly in critical domains such as healthcare (Oakden-Rayner et al., 2020) and social sciences (Dressel & Farid, 2018). The awareness of the potential negative consequences resulting from spurious correlations has captured significant attention within the machine learning community. Consequently, there has been substantial interest in developing strategies to address the impact (Sohoni et al., 2020; Sagawa et al., 2020; Nam et al., 2020; Liu et al., 2021; Zemel et al., 2013).

We acknowledge the gap in the existing research, which falls short in providing solutions from the perspective of the model itself, specifically addressing the question of whether spurious correlation can be overcome by applying a patch to the DNN itself. Our main objective in this work is to provide an understanding of DNNs in the context of spurious correlation. Specifically, we aim to address the following research questions:

(1) *What factors contribute to the reliance of DNNs on spurious features during training?* There is a common intuition that DNNs often demonstrate a tendency to achieve lower loss for easily learnable examples (i.e., samples whose labels can be inferred not only from task-relevant

features) while experiencing higher loss for more challenging examples during the early stages of training. Building upon this intuition, several methods have been developed to address bias in DNNs. While this intuitive phenomenon serves as a guiding principle for several methods (Yang et al., 2023a; Liu et al., 2021) aimed at mitigating bias in DNNs, the understanding of its occurrence has not been established in those works. To our knowledge, Adnan et al. (2022) is the first work that attempted to address this question through the information bottleneck framework. However, their discovery lacks finer granularity as they did not identify the specific factor(s) (e.g., complexity of the model, data distributions, optimization, etc.) that give rise to this bias phenomenon, leaving this research question unresolved. In this work, we would like to address this question through the lens of NTK, narrowing our investigation to the architecture and learning algorithm (gradient descent).

(2) *Can a procedural approach be developed to effectively tailor the DNN with the aim of enhancing the robustness?* Building upon the previous question, we would like to develop a novel approach grounded in deep learning principles that surpasses the constraints imposed by the conventional sample/feature paradigm (sample and feature paradigms are described in Section 2 below).

To answer those questions, we leverage the insights gained from relationship between neural networks and kernel machines – Neural Tangent Kernels (NTKs) (Jacot et al., 2018). The NTK is a concept in deep learning theory that characterizes the dynamics of learning when DNNs are trained using gradient descent. This kernel is formally defined as the expected product of gradients between two data points with respect to the weights initialization. The kernel matrix (also known as Gram matrix) adeptly compresses the dataset, model architecture and the learning algorithm (gradient descent) into a single compact representation (Shawe-Taylor & Cristianini, 2004). This compression allows us to exploit the classical framework of kernel methods to conduct comprehensive analysis of a DNN.

Our contributions can be summarized as follows:

- Our findings reveal that low-frequency kernel eigenvectors are associated with features that are inherently easier to learn and exhibit relatively stronger bias. When these features become entangled in spurious correlations with the target labels, it adversely affects the generalization capacity of DNNs.

- We introduce a novel approach that alleviates the impact of spurious correlations, all while keeping input features, training distribution, and loss function unchanged, and without requiring any knowledge of the spurious attributes.

The structure of the paper is outlined as follows: in Section 2, we delve into previous studies that bear relevance to our research questions. Following this, in Section 3, we introduce the notations and provide some background information. Subsequently, Section 4 covers the studies addressing research question (1), which aims to uncover the underlying reasons behind generalization issues caused by spurious correlations. Then, to address question (2), we proposed a solution in Section 5. Lastly, we discuss the limitation and future directions in Section 6. Additional results and experimental details can be found in the supplementary material.

## 2 RELATED WORK

Our work mainly involves three areas: subgroup robustness, spectral bias, and neural tangent kernels. The discussion of NTKs is provided in Appendix A.

**Subgroup robustness** Existing approaches can primarily be categorized into two main perspectives: *sample level* and *feature level* methods. The first perspective focuses on the *sample level*, taking into account the fact that poor generalization to the minority class stems from the insufficient contribution of samples from rare subgroups during empirical risk minimization (ERM). In this context, Sagawa et al. (2020) proposed GDRO, which optimizes the model directly with respect to the worst subgroup loss by leveraging full access to biased-attribute labels. On the other hand, approaches like Liu et al. (2021); Nam et al. (2020); Sohoni et al. (2020); Kim et al. (2023); Kamiran & Calders (2012) do not rely on biased-attribute labels but instead use proxies to identify rare samples and uplift their sample probability or loss. Additionally, Kirichenko et al. (2023) employs a balanced validation set to fine-tune the last layer of the DNN. The second perspective shifts its focus to the *feature level*, aiming to address the impact of spurious features by either eliminating them completely or diminishing

their influence. Zemel et al. (2013) and Arjovsky et al. (2020) explore the learning of alternative features representations guided by specific learning objectives, Yao et al. (2022) utilizes the mixup technique (Zhang et al., 2018) to mitigate the presence of the spurious feature, and Taghanaki et al. (2022) adopts a selective feature removal approach followed by fine-tuning of the DNN. Tiwari & Shenoy (2023) adjusts learned features by selectively fine-tuning subset of layers. In addition, contributions in the realm of spurious correlation analysis have been made by Adnan et al. (2022) and Yang et al. (2022b). In this work, we directly address the spurious correlation problem within the target model without relying on auxiliary networks, modifying the loss function, or having access to label information. It can be argued that approaches like Liu et al. (2021) which use a partially-trained DNN to identify underperforming subgroups then upweight them during, can be considered as addressing the problem to some extent from the model's perspective. However, it's important to highlight that these methods primarily concentrate on resolving the issue either at the individual sample level or the feature level.

**Simplicity bias & spectral bias**    It was shown in Brutzkus et al. (2017) that DNNs trained with stochastic gradient descent (SGD) possess an inductive bias towards linear interpolation for training examples. Similarly, Nakkiran et al. (2019) discovered a progressive learning process in theses networks, wherein they learn functions of growing complexity, initially capturing low-complexity (linear) representations and subsequently advancing towards high-complexity (non-linear) representations, this behaviour is known as *simplicity bias*. Another line of works (Rahaman et al., 2019; Cao et al., 2020; Xu et al., 2019; Xu, 2020) employed the principle of Fourier analysis to explore simplicity bias, with complexity being characterized in terms of frequency, commonly referred to as *spectral bias* or *frequency bias*. While beneficial in terms of robustness in certain contexts (Qian et al., 2020; Awasthi et al., 2020), simplicity bias can also be detrimental in other situations, such as domain adaptation where the simplistic representations do not faithfully represent the relevant features necessary for predictions in other domains, resulting in poor out-of-distribution performance (Shah et al., 2020). Yang & Salman (2020) established a connection between spectral bias and the spectrum of the NTK. They showed that the complexity of the eigenbases learned by the DNN is determined by the corresponding eigenvalues, specifically, smaller eigenvalues indicate more complex functions and vice versa. This aligns with result from Basri et al. (2019) that losses projected onto low-frequency target functions converge to zero at a higher rate than that of high-frequency target functions. This finding motivated several works aimed at accelerating the learning of higher-frequency target functions. Yang et al. (2022a); Tancik et al. (2020) leveraged random Fourier features (Rahimi & Recht, 2007) to widen the spectrum. Yu et al. (2023) uses the Sobolev norm to adjust the priority of learning functions with specific bandwidths. Our work aims to leverage the phenomenon of spectral bias to fundamentally understand the impact of spurious correlation to DNNs. Specifically, we hypothesize that low-frequency components in the functional space are associated to spurious features; as a consequence of spurious correlations, the model latches onto these low-frequency components, impeding its ability to learn more complex features which are more predictive beyond the training environment.

## 3 PRELIMINARIES

In this section, we provide a brief overview of the background and mathematical notation relevant to our work. We use lowercase bold letters to denote vectors (e.g., $\mathbf{y} = (y_1, \ldots, y_n)$ for the training labels) and uppercase bold letters for the matrices (e.g., $\mathbf{X} = (x_1, \ldots, x_n)^\top \in \mathbb{R}^{n \times d}$ where $x_i \in \mathbb{R}^d$ represent the complete training set with $n$ samples), with $(x, y)$ denoting individual data samples from the dataset. A DNN is defined as a function $f_\theta : \mathcal{X} \to \mathcal{Y}$, parameterized by $\theta$, and the loss function is defined as $\mathcal{L} : \mathcal{Y} \times \mathcal{Y} \to \mathbb{R}$. In this context, $\mathcal{X}$ represents the input space, and $\mathcal{Y}$ represents the output space. We write $f(\mathbf{X}) = (f(x_1), \ldots, f(x_n))$ as the vectorization over $n$ samples. Additionally, we introduce $\dot{f}$ to denote the time derivative of $f$.

### 3.1 SPURIOUS CORRELATION

We consider the setting where the input space is composed of two distinct feature spaces: $\mathcal{X} := \mathcal{X}_y \times \mathcal{X}_s$, where $\mathcal{X}_y$ denotes the invariant feature space which is exclusively relevant to the task. On the other hand, $\mathcal{X}_s$ denotes the irrelevant feature space associated with certain attribute(s) (e.g.,

colour, background)[1]. The composition exhibits in many forms such as overlapping, superposition or concatenation of $\mathcal{X}_y$ and $\mathcal{X}_s$ (Fig. 1 showcases several variants of the MNIST dataset for studying spurious correlations). We use $s \in \mathcal{S}$ to denote the label for the spurious attribute while the subgroups are represented by $g \in \mathcal{G} := \mathcal{Y} \times \mathcal{S}$. Formally, spurious correlation refers to the scenario where a statistical dependency between the target $Y$ and the attribute $S$ is observed solely within the training samples:

$$\mathbb{P}_{\text{train}}(X, Y, S) \propto \mathbb{P}_{\text{train}}(Y|S) \, \mathbb{P}_{\text{train}}(S) \tag{1}$$

$$\mathbb{P}_{\text{test}}(X, Y, S) \propto \mathbb{P}_{\text{test}}(Y) \, \mathbb{P}_{\text{test}}(S) \tag{2}$$

where in the training set the target is entangled with the bias attribute $\mathbb{P}_{\text{train}}(Y|S) \neq \mathbb{P}_{\text{train}}(Y)$. However, this entanglement is either absent or significantly weakened when considering the test environment. When machine learning models are trained on data with spurious correlations, they may mistakenly learn to rely on these correlations instead of capturing the underlying true patterns. Consequently, the models fail to generalize well to unseen data, leading to poor performance and inaccurate predictions. Furthermore, if the spurious correlations align with sensitive information such as race or gender, the models may inadvertently learn and propagate societal biases, leading to unfair and discriminatory outcomes. Given this challenge, it is imperative to develop machine learning models that exhibit robustness by effectively distinguishing between genuine correlations and spurious correlations. By doing so, the adverse effects of spurious correlations can be mitigated, and the robustness of the models can be improved.

We employ two key performance metrics in our evaluation. The first metric is the *average accuracy*: $\mathbb{E}_{(x,y)}\left[\mathbb{1}\left[f(x) = y\right]\right]$, which provides an assessment of the overall model performance. The second metric is the *worst-group accuracy* (Sagawa et al., 2020) defined as the 'accuracy' of the worst-performing subgroup: $\min_{g' \in \mathcal{G}} \mathbb{E}_{(x,y)|g=g'}\left[\mathbb{1}\left[f(x) = y\right]\right]$. More formally, the worst-group accuracy is known as the *worst true positive rate of one group versus the other groups*. This is a commonly used evaluation measure in the field of subgroup robustness (Idrissi et al., 2022; Yang et al., 2023b).

## 3.2 Neural Tangent Kernel

DNNs are commonly trained using gradient descent, following the gradient flow:

$$\dot{\theta}_t := \theta_{t+1} \leftarrow \theta_t - \eta \nabla_{\theta_t} \mathcal{L}\left(f(\mathbf{X}), \mathbf{y}\right) \tag{3}$$

where $\eta$ is the learning rate. For the sake of simplicity, moving forward we will omit the argument in $\mathcal{L}$. By applying the chain rule, we can derive the training evolution of $f$:

$$\begin{aligned} \dot{f}_t(\mathbf{X}) &= \nabla_{\theta_t} f(\mathbf{X})\, \dot{\theta}_t \\ &= -\eta \nabla_{\theta_t} f(\mathbf{X}) \nabla_{\theta_t} f(\mathbf{X})^\top \nabla_{f(\mathbf{X})} \mathcal{L}. \end{aligned} \tag{4}$$

The NTK is defined as the product of gradients of the DNN with respect to its outputs evaluated with $\theta$ at time $t$: $\kappa_{\theta_t}(x, x') = \nabla_{\theta_t} f(x) \nabla_{\theta_t} f(x')^\top$. Thus, the expression for the training evolution can be written as follows:

$$\dot{f}_t(\mathbf{X}) = -\eta \kappa_{\theta_t}\left(\mathbf{X}, \mathbf{X}\right) \nabla_{f(\mathbf{X})} \mathcal{L}. \tag{5}$$

Under the infinite-width assumption (Jacot et al., 2018), the training falls in the lazy regime (Chizat et al., 2020) where the NTK converges to a deterministic kernel at initialization $\kappa_{\theta_t} \to \kappa_{\theta_0}$. In other words, the NTK remains constant at the initialization, enabling us to predict the DNN's behaviour *a priori* without running gradient descent. Again, when the context is clear, we omit the subscript of $\kappa$ for simplicity and use $\mathbf{H} := \kappa\left(\mathbf{X}, \mathbf{X}\right) \in \mathbb{R}^{n \times n}$ to denote the Gram matrix of the training set. Similarly, the evolution of $f$ on an individual sample $x$ can be described by the following ODE:

$$\dot{f}_t(x) = -\kappa\left(x, \mathbf{X}\right) \nabla_{f(\mathbf{X})} \mathcal{L}. \tag{6}$$

When an L2 loss $\mathcal{L}(\hat{y}, y) = \frac{1}{2}\|\hat{y} - y\|^2$ is used, the ODEs can be solved analytically (Lee et al., 2019),

$$f_t(\mathbf{X}) = \left(I - e^{-\eta \mathbf{H} t}\right) \mathbf{y} + e^{-\eta \mathbf{H} t} f_0(\mathbf{X}) \tag{7}$$

where $f_0$ is the initial condition. In the asymptotic regime, as the training progresses indefinitely, the evolution of $f$ converges towards a kernel machine

$$f(x) \stackrel{t \to \infty}{\approx} \kappa(x, \mathbf{X}) \mathbf{H}^{-1} \mathbf{y}. \tag{8}$$

---

[1]It is important to note that we make no assumptions about the spurious attribute(s). The framework presented is not restricted to a single attribute and can be extended to incorporate multiple ones (e.g., race, gender, etc.) (Wiles et al., 2021).

# 4 UNDERSTANDING SPURIOUS FEATURE RELIANCE IN DNNs: SPECTRAL INSIGHTS

In this section, we present an empirical analysis examining how spurious correlations impact DNNs. The experimental details can be found in Appendix B.

## 4.1 DATASETS

We extend the datasets proposed in Kirichenko et al. (2023) and Taghanaki et al. (2022) by introducing additional variations and complexities. The dataset details are as follows: for every dataset, within the training set, the negative class is assigned a specific spurious attribute with a probability of $\alpha$, while the positive class is assigned the same attribute with a probability of $1 - \alpha$, where $\alpha \in [0, 0.5]$ (details can be found in Appendix B). In the test set, we use $\alpha = 0.5$ to replicate a scenario where spurious correlation is absent. We use *bias-aligned* to refer to samples from frequently represented subgroups in the training set, and *bias-conflicting* for samples from rarely observed subgroups.

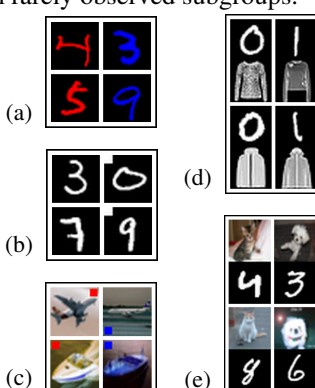

**CMNIST** (Fig. 1a): we establish binary classes by assigning negative labels to digits 0-4 and positive labels to digits 5-9. The spurious attribute is the foreground color: red and blue.

**Biased-MNIST** (Fig. 1b): similar to the setup in **CMNIST** but with an additional white patch serves as the spurious attributes.

**Biased-CIFAR** (Fig. 1c): we use {airplanes,ships} as target classes. A colored patch (red or blue) randomly appears in the corners as the spurious attribute.

**Fashion-MNIST** (Fig. 1d): we use {pullovers,coats} as target classes with digits zero and one as spurious attributes.

**CIFAR-MNIST** (Fig. 1e): the same task as **CMNIST** but with cats and dogs as spurious attributes.

Figure 1: Datasets with spurious features.

## 4.2 DECOMPOSITION OF $f(x)$

Since the kernel matrix $\mathbf{H}$ for the training set is positive definite, it can be factorized as $\mathbf{H} = \mathbf{U}\mathbf{\Lambda}\mathbf{U}^\top$, where $\mathbf{U} = (\mathbf{u}_1, \ldots, \mathbf{u}_n)$ is an orthogonal matrix whose columns are the eigenvectors of $\mathbf{H}$, and $\mathbf{\Lambda} = \text{diag}\{\lambda_1, \ldots, \lambda_n\}$ with $\lambda_1 \geqslant \lambda_2 \geqslant \cdots \geqslant \lambda_n$ is a diagonal matrix containing the corresponding eigenvalues in decreasing order. Then, we can rewrite Eq. (7) as

$$f_t(\mathbf{X}) - \mathbf{y} = \mathbf{U}e^{-\eta\Lambda t}\mathbf{U}^\top \left(f_0(\mathbf{X}) - \mathbf{y}\right), \tag{9}$$

by applying a change of basis, we obtain

$$\mathbf{U}^\top \left(f_t(\mathbf{X}) - \mathbf{y}\right) = e^{-\eta\Lambda t}\mathbf{U}^\top \left(f_0(\mathbf{X}) - \mathbf{y}\right). \tag{10}$$

The above expression implies that the update during training is performed along the directions defined by the eigenbasis where the magnitudes are scaled by the corresponding eigenvalues. In other words, each basis function in $\mathbf{U}$ converges with an exponential decay rate of $\eta\lambda_i t$.

Based on the decomposition in Eq. (10), training a DNN with gradient descent is akin to sequentially fitting multiple functions, where the rate of fitting is governed by the associated eigenvalues. Moreover, we can interpret the eigenfunctions as distinct hypotheses that serve to separate examples in the input space. These hypotheses can manifest in various forms, with some relying on low-level features while others depend on higher-level features (Tsilivis & Kempe, 2022). Furthermore, we can establish a connection between this interpretation and the well-known fact that DNNs tend to exhibit a strong reliance on spurious attributes during the early stages of training, which prompts the following question: *given that eigenfunctions with large eigenvalues are fitted more rapidly, does this imply that*

---

[2]The first eigenfunction $f_1(x)$, being a constant classifier, is excluded as its gradient is directly proportional to the input.

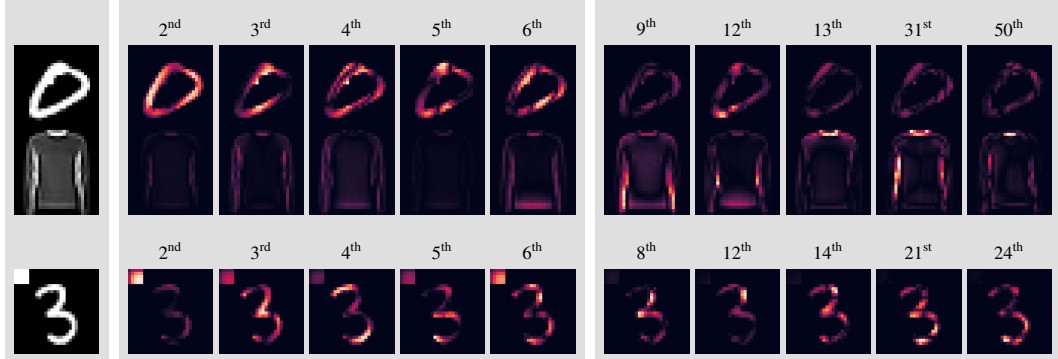

Figure 2: Input images are shown on the left, the middle column displays saliency maps for various eigenfunctions with higher activation in the spurious feature region, and the right column displays the core feature region. The indices of the $f_i(x)$ are displayed on the top[2]. Top row: `Fashion-MNIST` dataset; bottom row: `Biased-MNIST` dataset. The `Biased-CIFAR` example is illustrated in Fig. C.3.

*they are inherently tied to the spurious attributes?* The prediction score made by $f(x)$ (in Eq. (8)) (e.g., as a binary classification score ranging from zero to one) for a given sample $x$ is obtained by a weighted (by eigenvalues) sum of scores given by multiple unique[3] functions $f_i(x)$. That is:

$$f(x) = \sum_{i=1}^{n} f_i(x) \;,\; f_i(x) = \frac{1}{\lambda_i} \kappa(x, \mathbf{X})(\mathbf{u}_i \otimes \mathbf{u}_i)\mathbf{y}. \tag{11}$$

We can interpret $f_i(x)$ as functions that correspond to specific features in the input space. To understand the role of individual eigenfunctions, we evaluate the influence of the input on the prediction for each $f_i(x)$ by computing the derivative of the loss with respect to the input: $\nabla_x \mathcal{L}(f_i(x), y)$. This gradient map, also known as the *saliency map* (Simonyan et al., 2014), is a standard attribution technique used to interpret the importance of features for DNNs in making predictions. The saliency maps presented in Fig. 2 highlight two categories: saliency maps for $f_i(x)$ that rely on spurious features $\mathcal{X}_s$ and saliency maps for $f_i(x)$ that rely on core features $\mathcal{X}_y$. We observe that the lower-order eigenfunctions tend to exhibit a greater reliance on the spurious features, while the higher-order eigenfunctions demonstrate a stronger dependence on the core features.

## 4.3 FEATURE COMPLEXITY

One quantitative method for assessing feature complexity is by evaluating the number of eigenbases required to model the training samples. In this context, simpler features demand fewer basis functions for an accurate fit, while more complex features tend to necessitate a larger number of basis functions to effectively capture the data.

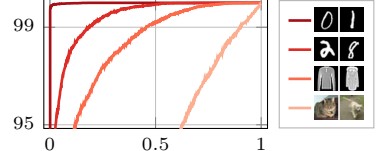

Figure 3: Training accuracy of $\mathbf{H}_k$. $x$-axis: normalized index $k$.

Fig. 3 visualizes the performance when considering a subset of eigenbases (or a truncated spectrum) i.e., $\mathbf{H}_k = (\mathbf{u}_1, \dots, \mathbf{u}_k) \times \text{diag}\{\lambda_1, \dots, \lambda_k\} \times (\mathbf{u}_1, \dots, \mathbf{u}_k)^\top$. We observe that the simplest feature, distinguishing between digit zero and digit one (`ZeroOne`), achieves 100% training accuracy with just a few of the top eigenbases. As the complexity of the features increases, such as in the case of binarized MNIST, approximately the top 20% of the basis functions are needed to attain a 99% accuracy rate. For distinguishing between pullover and coat (`PulloverCoat`), the top 30% is required, and in the case of distinguishing between cats and dogs, the top 90% are necessary.

This measure highlights the potential challenge in learning the task when the feature domain consists of multiple features that vary significantly in complexity. For instance, the feature of dataset `Fashion-MNIST` is composed of `ZeroOne` and `PulloverCoat`. Subjected to spurious correlations, we observe a decline in performance especially the worst-group performance (in Table C.2) when

---

[3]The uniqueness stems from the orthogonality of the eigenvectors.

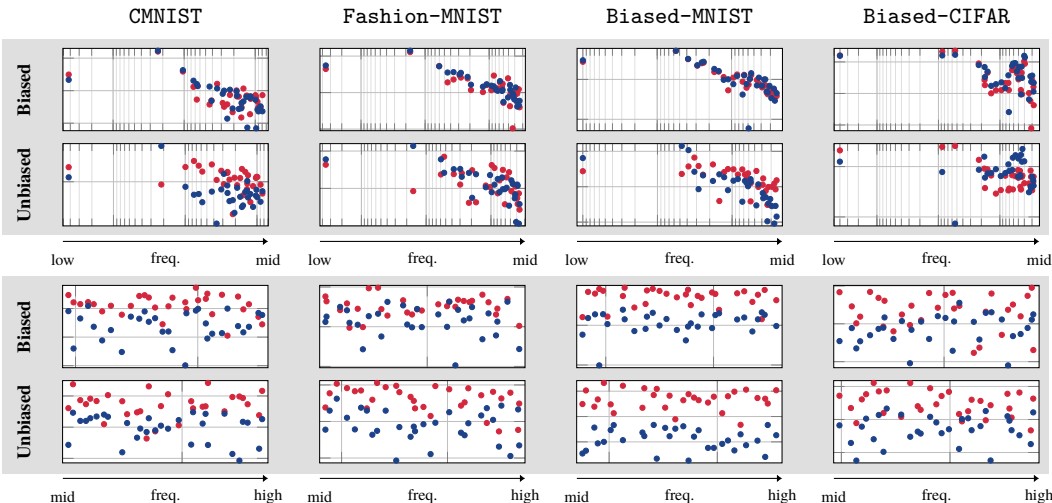

Figure 4: Alignment $A_i(\mathbf{M})$ ($y$-axis) of eigenvectors and target labels $\mathbf{y}$ (red dots), and alignment of eigenvectors and bias labels $\mathbf{s}$ (blue dots) is shown across frequency spectrums ($x$-axis) on both biased and unbiased datasets. Top row: low-frequency to mid-frequency, bottom row: mid-frequency to high-frequency. Only on the unbiased dataset, the eigenvectors from the low-frequency band exhibit a higher alignment with the target (red dots), particularly in the cases of `CMNIST` and `Biased-MNIST`. We attribute this to the fact that the spurious features being considerably simpler compared to the core feature. However, the components from the mid-frequency band consistently display a stronger alignment with the target, irrespective of the presence of spurious correlation.

the feature complexity of the target (`PulloverCoat`) exceeds that of the bias (`ZeroOne`). However, when the situation is reversed (with `ZeroOne` as the target and `PulloverCoat` as the bias), we observed no generalization issue. In a more extreme scenario where the bias is completely correlated with the target ($\alpha = 0$) on `CIFAR-MNIST`, there are no generalization issues until roles are reversed. These findings provide an insight into our research question by revealing that poor robustness can be attributed not only to statistical bias (spurious correlations) in the dataset but also to discrepancies in feature complexity.

**Alignment** Here, we measure the alignment between the Gram matrix $\mathbf{H}$ and the target labels $\mathbf{y}$, as well as between the spurious labels $\mathbf{s}$ (Cristianini et al., 2001):

$$A(\mathbf{M}) = \frac{\langle \mathbf{H}, \mathbf{M} \rangle_{\mathbf{F}}}{\sqrt{\langle \mathbf{H}, \mathbf{H} \rangle_{\mathbf{F}} \langle \mathbf{M}, \mathbf{M} \rangle_{\mathbf{F}}}} \quad (12)$$

where $\mathbf{M} \in \{\mathbf{y}\mathbf{y}^\top, \mathbf{s}\mathbf{s}^\top\}$. Using the decomposition of $\mathbf{H}$, we can express the alignment as a sum of contributions from individual eigenvectors:

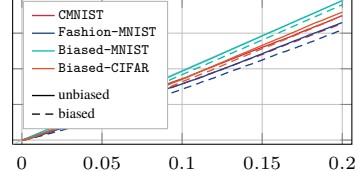

Figure 5: The $y$-axis is the function $\frac{1}{k} \sum_{j=1}^{k} \mathbb{1}\left[A_i(\mathbf{y}\mathbf{y}^\top) > A_i(\mathbf{s}\mathbf{s}^\top)\right]$ and the $x$-axis is $k$, the normalized frequency index.

$$A(\mathbf{M}) = \sum_{i=1}^{n} A_i(\mathbf{M}) \ , \ A_i(\mathbf{M}) = \frac{\lambda_i \langle \mathbf{u}_i \otimes \mathbf{u}_i, \mathbf{M} \rangle_{\mathbf{F}}}{\sqrt{\sum_{i=1}^{n} \lambda_i} \sqrt{\langle \mathbf{M}, \mathbf{M} \rangle_{\mathbf{F}}}} \quad (13)$$

The alignment quantity can be used as a proxy for evaluating the extent to which each target function, derived from the eigenbasis, effectively captures the label information. In other words, by measuring the relative angle between each component and the training labels, we can assess their contribution to the overall loss, as defined in Eq. (10). As observed in Fig. 4, low-order components (eigenvectors with larger eigenvalues) are more strongly associated with spurious labels $\mathbf{s}$ compared to target labels $\mathbf{y}$ when there is a presence of spurious correlation in the dataset.

More specifically, Fig. 5 shows the (normalized) overall alignment when considering the top $k$ (sorted by descending order of associated eigenvalues) eigenbases. The slope indicates the extent to which

the top $k$ components align with $\mathbf{y}$ rather than $\mathbf{s}$. The slopes for unbiased datasets are consistently steeper compared to that of biased datasets which implies that components associated with strong bias (characterized by larger eigenvalues) tend to align more with $\mathbf{s}$ in the biased scenarios, indicating that DNNs rely more on the bias attribute in the presence of spurious correlation.

## 5 ADDRESSING SPURIOUS FEATURE RELIANCE WITH SPECTRUM MODIFICATION

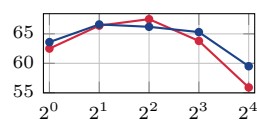

Figure 6: Worst-group accuracy with varying depth on CMNIST and Fashion-MNIST.

Ideally, our aim is to design a network architecture that is immune to spurious corrections. However, in practice, the design is challenged by a lack of well-defined principles, primarily due to our limited understanding of which specific neural architecture can provide effective solutions. Our empirical analysis reveals that spurious features are prioritized in learning and inference due to their simplicity. Furthermore, the strong correlation between spurious attributes and target labels in the training data leads to a stronger alignment, causing predictions to focus on spurious features and resulting in poor generalization. As visually depicted in Fig. C.1, shallow (underparameterized) networks exhibit narrower spectra, while deeper (overparameterized) networks showcase broader spectra. Illustrated in Fig. 6, there is a non-monotonic trend indicating that increasing network depth (or increasing spectral width) enhances subgroup robustness but starts deteriorating after reaching an optimal depth. This observation aligns with the findings of Yang & Salman (2020), highlighting that while deeper networks can learn more complex features, excessive depth may lead to deterioration in performance.

Moreover, based on our discovery that eigenfunctions associated with $\mathcal{X}_s$ and $\mathcal{X}_y$ lie within different spectral ranges, a strategy to mitigate spurious correlation involves raising the values of $\lambda_i$ for eigenfunctions associated with $\mathcal{X}_y$ which essentially results in a wider spectrum. The duality of kernel states that stretching the spectrum of a kernel corresponds to shrinking the kernel in the spatial space, thereby enhancing the capability to capture high-frequency features. As the spectrum widens, the kernel tends to the behaviour of the Dirac delta function such that two points are considered close only when they possess finely detailed (high-frequency) features in common. Consequently, a kernel with an extensive spectrum tends to incorporate noisy features resulting in a diagonally dominant kernel causing overfitting. With these observations, we seek to approach this issue from a different angle – specifically, *can we reverse engineer a kernel that promotes high generalization to uncover the corresponding neural architecture?* However, directly constructing such a kernel can be practically challenging and may require heuristic computations on the holdout dataset. Instead, we adopt an alternative approach to construct a new kernel by manipulating the kernel spectrum. Given that $\kappa(x, x') = \sum_{i=1}^{\infty} \lambda_i \phi_i(x) \phi_i(x')$, where $\phi$ is the eigenfunction of $\kappa$, we introduce a new kernel $\tilde{\kappa}$ from the eigenspace of $\kappa$ by modifying the eigenvalues: $\tilde{\kappa}(x, x') = \sum_{i=1}^{\infty} \nu(\lambda_i) \phi_i(x) \phi_i(x')$, with $\nu : \mathbb{R} \to \mathbb{R}_{>0}$ ensuring positive definiteness. Empirically, we can modify the existing Gram matrix $\mathbf{H}$:

$$\tilde{\kappa}(\mathbf{X}, \mathbf{X}) = \widetilde{\mathbf{H}} = \mathbf{U}\widetilde{\Lambda}\mathbf{U}^{\top} \tag{14}$$

where $\widetilde{\Lambda} = \mathrm{diag}\{\nu(\lambda_1), \ldots, \nu(\lambda_n)\}$. This approach can be interpreted as changing the spectral characteristics of NTK. By tuning the spectrum $\lambda$ via $\nu$, we aim to strike a better balance between core and spurious features, ultimately improving the generalization of the model. Here we use the following $\nu$:

$$\nu(\lambda_i) = \lambda_i \left(e^{-\gamma i} + \beta\right) \tag{15}$$

where $i \in \left[\frac{1}{n}, 1\right]$ is the normalized frequency index, and the parameters $\gamma > 0$ and $\beta \geq 0$ determine the spectrum of $\tilde{k}$. Original eigenvalues $\lambda$ are scaled by the factor $\left(e^{-\gamma i} + \beta\right)$ which is greater than 1 when $\beta > 0$ for low-order components (small $i$) and tends to $\beta$ for high-order components (larger $i$), while $\gamma$ controls the decay rate of the eigenvalues, in other words, the shape of the spectrum.

Table 1 highlights a substantial enhancement in generalization performance through the modification of the kernel spectrum. We observed a significant decrease in the average-worst performance gap $\Delta$ coinciding with an increase in overall average performance. This improvement underscores the potential of tuning the spectrum to guide the model away from learning overly simple features that could otherwise impair overall performance. Moreover, Fig. 7 illustrates the dynamics of the

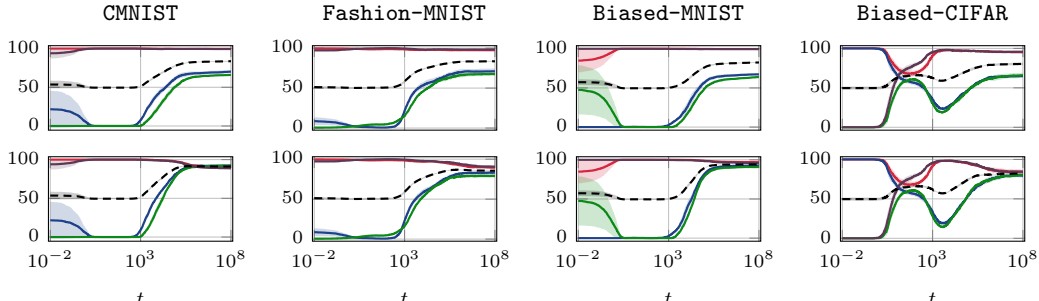

Figure 7: Average accuracy (dashed line) and the accuracy for every subgroup (four color lines) with original NTK $\kappa$ (top row) and the modified NTK $\tilde{\kappa}$ (bottom row). The $x$-axis corresponds to the time in the ODE Eq. (7). Subgroups with aligned biases (red and purple lines) achieved faster optimal performance than those with conflicting biases (blue and green lines). After spectrum modification, all subgroups converged to similar performance levels.

performance. Subgroups with aligned biases reached optimal performance quicker than those with conflicting biases, while subgroups with conflicting biases eventually converged to performance levels below the average. However, following the spectrum modification, all subgroups converged to similar performance levels. The choice of hyperparameter selection for $\gamma$ and $\beta$ is discussed in Appendix C.1.

| | $\kappa$ | | | $\widetilde{\kappa}$ | | |
|---|---|---|---|---|---|---|
| | avg. | worst | $\Delta$ | avg. | worst | $\Delta$ |
| CMNIST | $83.8_{\pm 0.4}$ | $67.5_{\pm 0.7}$ | $16.3_{\pm 0.4}$ | $93.1_{\pm 0.5}$ | $88.5_{\pm 1.3}$ | $4.6_{\pm 0.8}$ |
| Fashion-MNIST | $83.2_{\pm 0.9}$ | $66.2_{\pm 2.6}$ | $16.9_{\pm 1.9}$ | $85.1_{\pm 0.2}$ | $76.8_{\pm 1.9}$ | $8.3_{\pm 1.8}$ |
| Biased-MNIST | $83.8_{\pm 1.8}$ | $67.1_{\pm 3.8}$ | $16.7_{\pm 2.1}$ | $96.9_{\pm 0.4}$ | $95.9_{\pm 0.9}$ | $1.0_{\pm 0.6}$ |
| Biased-CIFAR | $80.6_{\pm 1.4}$ | $64.6_{\pm 1.2}$ | $16.0_{\pm 0.9}$ | $83.0_{\pm 1.5}$ | $75.9_{\pm 1.0}$ | $7.1_{\pm 0.6}$ |

Table 1: Performance of NTK $\kappa$ and NTK with modified spectrum $\tilde{\kappa}$ on various datasets. Here, $\Delta$ corresponds to the gap between the average performance and the worst-group performance (a smaller gap indicates that the model exhibits less preference for any specific subgroups).

## 6 CONCLUSIONS AND FUTURE WORK

In this work, we establish a fundamental connection between the phenomenon of spectral bias in DNNs and subgroup robustness. In the first part of our studies, we discovered that spurious correlations within the dataset negatively impact DNN generalization only when the bias attributes' complexity significantly lags behind that of the core attributes. Specifically, we demonstrate that low-frequency components correspond to simplistic features. When these simplistic features become entangled with the target, despite lacking predictive power, DNNs will overly rely on them during inference.

Building upon this insight, we introduce a novel approach involving the modification of the NTK spectrum to address the subgroup robustness issue. We empirically show that this modification effectively guides DNNs to bypass simplistic features, thereby improving the robustness. This approach solely alters network properties, eliminating the requirement for knowledge of spurious attributes, auxiliary networks, specific loss functions, or the privilege of changing training samples. One shortcoming of the method is the computational constraint: constructing a Gram matrix requires $O(n^2)$ operations and the decomposition requires $O(n^3)$ operations, which can be a significant challenge for larger datasets. In future work, one could explore more practical approaches to adapt the modification of network spectra, such as controlling the spectrum during the feature learning process (Tancik et al., 2020; Tiwari & Shenoy, 2023).

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

Supplementary material

This is the supplementary material for the paper titled "Understanding and addressing spurious correlation via Neural Tangent Kernels: A spectral bias perspective". Section A provides additional discussion on the NTK. Further information regarding the experiments, including details on the inference method, network architecture, and datasets, can be found in Section B. Additional results are presented in Section C, while the process of hyperparameter selection is discussed in Section C.1. Results related to the dynamical aspect are provided in Section C.2 and Section C.3, and Section C.4 provides results for finite networks trained with SGD.

## A Discussion on neural tangent kernels

The NTK have emerged as a significant area of research within the deep learning community. Initially introduced by Jacot et al. (2018), NTKs provide a mathematical framework to analyze the behavior of deep neural networks during training. On the theoretical front, works such as Lee et al. (2019) have explored the NTK's connection to the infinite-width limit of neural networks, shedding light on the linear behavior of these models. Additionally, several studies have investigated the role of NTKs in understanding optimization dynamics during training (Arora et al., 2019a; Chizat et al., 2020; Cao & Gu, 2019; Arora et al., 2022). Furthermore, Arora et al. (2019b) has extended the application of NTKs to convolutional neural networks (CNNs), expanding their relevance to various deep learning architectures. Overall, the growing body of work on NTKs underscores their significance in advancing our understanding of deep learning theory and enhancing practical training methods. Our study utilizes this framework to address spurious feature reliance in deep learning

## B Experimental details

For the stationary setup, we follow a similar experimental setup to that described in Lee et al. (2020), where we compute the prediction using the exact inference (in Eq. (8)) which corresponds to the mean prediction of infinitely many ensembles. While in the dynamical setup, we compute the prediction by solving the ODE (in Eq. (6)). The finite-width network is trained using a SGD optimizer with a fixed learning rate of $10^{-3}$ and a momentum of 0.9 for 10,000 training steps.

**Architectures**  By default, we use the standard parametrization (Sohl-Dickstein et al., 2020), the rectified linear unit (ReLU) non-linearity and initialization with variance $\sigma_W^2 = 2$ and $\sigma_b^2 = 0.1$. The abbreviation "CNN" refers to convolutional neural networks, with the number indicating the depth of the network. All experiments were conducted using the Neural Tangents library (Novak et al., 2020) and JAX (Bradbury et al., 2018). The finite-width network architecture (in Section C.4) consists of 4 blocks of {`Conv2d`, `BatchNorm2d`, `ReLU`, `MaxPool2d`} as the backbone and one linear layer as the classifier.

**Datasets**  For all datasets, the training set consists of 10,000 samples, while both the test and validation sets consist of 2,000 samples, with the exception of `Biased-CIFAR`, which contains 1,000 samples in each set. All datasets are constructed using various existing datasets (LeCun & Cortes, 2010; Xiao et al., 2017; Krizhevsky & Hinton, 2009). We used different values of $\alpha$ for each dataset: $\alpha = 0.05$ for `CMNIST`, $\alpha = 0.1$ for `Fashion-MNIST`, $\alpha = 0.03$ for `Biased-MNIST`, and $\alpha = 0.2$ for `Biased-Fashion`. The samples are normalized to the range $[0, 1]$ before being fed into the network.

## C Additional results

To assess the reliance of predictions on spurious attributes, we conduct an experiment where we manipulate the training labels during the inference process. In Table C.2 we report the average accuracy and the worst group accuracy of the NTKs on the unbiased test set across different datasets. Initially, we use the target classes $Y$ as the training labels and keep the spurious attributes $S$ unchanged. This configuration resulted in a substantial discrepancy between the average performance and the worst group performance. However, when we reverse the roles and use the spurious attributes as the training labels while assigning the target classes as spurious attributes, we observe a significant

reduction in the performance gap, indicating a strong reliance on spurious attributes. Our NTK results are consistent with the findings presented in Nam et al. (2020), demonstrating that by appropriately choosing the attributes for $Y$ and $S$, the degradation of the worst group performance can be alleviated, and on all synthetic datasets, the degradation is entirely eliminated.

| Dataset | Target | Bias | Biased | | | Unbiased | | |
|---|---|---|---|---|---|---|---|---|
| | | | Avg. | Worst | $\Delta$ | Avg. | Worst | $\Delta$ |
| CMNIST | digit | color | $83.8_{\pm 0.4}$ | $67.5_{\pm 0.7}$ | $16.3_{\pm 0.4}$ | $97.0_{\pm 0.3}$ | $96.1_{\pm 0.6}$ | $0.8_{\pm 0.3}$ |
| | color | digit | $100.0_{\pm 0.0}$ | $100.0_{\pm 0.0}$ | $0.0_{\pm 0.0}$ | $100.0_{\pm 0.0}$ | $100.0_{\pm 0.0}$ | $0.0_{\pm 0.0}$ |
| Fashion-MNIST | fashion | digit | $83.2_{\pm 0.9}$ | $66.2_{\pm 2.6}$ | $16.9_{\pm 1.9}$ | $91.8_{\pm 0.7}$ | $90.3_{\pm 1.2}$ | $1.6_{\pm 0.6}$ |
| | digit | fashion | $100.0_{\pm 0.0}$ | $100.0_{\pm 0.0}$ | $0.0_{\pm 0.0}$ | $100.0_{\pm 0.0}$ | $99.0_{\pm 0.1}$ | $0.0_{\pm 0.1}$ |
| Biased-MNIST | digit | patch | $83.8_{\pm 1.8}$ | $67.1_{\pm 3.8}$ | $16.7_{\pm 2.1}$ | $97.7_{\pm 0.4}$ | $96.9_{\pm 0.5}$ | $0.8_{\pm 0.2}$ |
| | patch | digit | $100.0_{\pm 0.0}$ | $100.0_{\pm 0.0}$ | $0.0_{\pm 0.0}$ | $100.0_{\pm 0.0}$ | $100.0_{\pm 0.0}$ | $0.0_{\pm 0.0}$ |
| Biased-CIFAR | object | color patch | $80.6_{\pm 1.4}$ | $64.6_{\pm 1.2}$ | $16.0_{\pm 0.9}$ | $86.3_{\pm 1.0}$ | $83.7_{\pm 0.7}$ | $2.6_{\pm 0.5}$ |
| | color patch | object | $100.0_{\pm 0.0}$ | $100.0_{\pm 0.0}$ | $0.0_{\pm 0.0}$ | $100.0_{\pm 0.0}$ | $100.0_{\pm 0.0}$ | $0.0_{\pm 0.0}$ |

Table C.2: Performance of NTK across multiple datasets, considering the presence or absence of spurious correlations in the training set, and varying the target and bias.

**Characteristics of NTKs in relation to group robustness**    One aspect we investigate is the gradient similarities among subgroups. As shown in Fig. C.1, we observe that subgroups having the same spurious attributes exhibit higher gradient similarities than that of subgroups belonging to the same target classes. This suggests that DNNs place greater emphasis on the (dis)similarity of examples based on the spurious features rather than the core features. As the depth of DNNs increases, the distinguishability of samples in terms of gradients becomes less discernible which explains the observed drop in performance. Another explanation can be drawn from the spectrum (Fig. C.1, right) where deeper networks induce wider spectrum which capture wider input bandwidths (Tancik et al., 2020; Yang et al., 2022a). Consequently, the poor generalization performance can be attributed to the fitting of high-frequency features (at the tail of spectrum), which often contain noise.

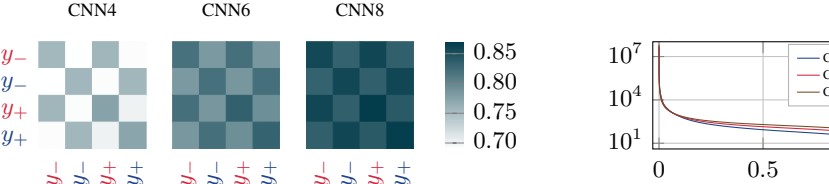

Figure C.1: *Left:* Gradient similarity over different subgroups. The subscript of $y$ represents the target class ("$-$" denotes digits $< 5$ and "$+$" otherwise) and the color represents spurious class (red and blue). Despite not belonging to the same target class, the spuriously correlated subgroups exhibit high gradient similarity. *Right:* kernel spectra, the width grows as the depth increases.

Figure C.2: Test accuracy (average) of minority subgroups with mixture of core and spurious eigenfunctions. We rank the relevance of eigenfunctions with respect to the core attributes based on the alignment difference between the target labels and the spurious labels, denoted as $A_i(\mathbf{yy}^\top) - A_i(\mathbf{ss}^\top)$. We observe a proportional deterioration in performance as the number of spurious eigenfunctions used in the prediction increases. The saliency maps for core and spurious eigenfunctions based on the alignment difference are displayed in Fig. C.5.

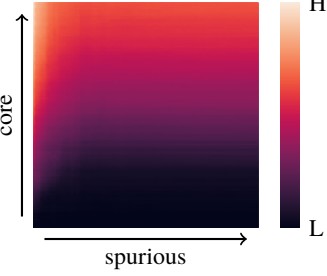

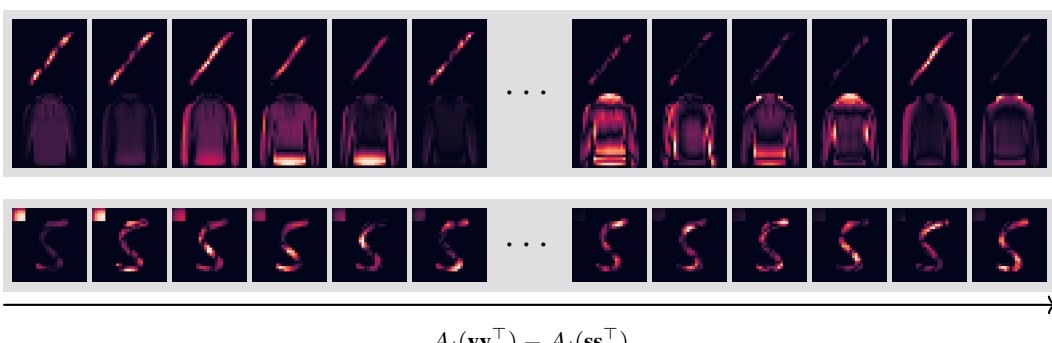

Figure C.3: Additional visualizations complementing Fig. 2 for the `Biased-CIFAR` dataset. Note that there are certain low-frequency components, specifically those at 5 and 7, exhibit higher activation within the core feature region. This is due to the overlapping feature complexity between $\mathcal{X}_x$ and $\mathcal{X}_s$, which explains the entanglement in the alignment with respect to $\mathbf{y}$ and $\mathbf{s}$ within the low-frequency spectrum in Fig. 4.

Figure C.4: Performance on `Cifar-MNIST` with a complete correlation ($\alpha = 0$) between the target and the bias. No deterioration in performance when the feature complexity of bias is larger than that of the target.

| Target | Bias | Avg. | Worst | $\Delta$ |
|---|---|---|---|---|
| digit | animal | $96.1_{\pm 0.6}$ | $93.6_{\pm 0.9}$ | $2.6_{\pm 0.4}$ |
| animal | digit | $52.1_{\pm 0.9}$ | $5.0_{\pm 1.0}$ | $47.0_{\pm 0.8}$ |

$$A_i(\mathbf{y}\mathbf{y}^\top) - A_i(\mathbf{s}\mathbf{s}^\top)$$

Figure C.5: Saliency maps of eigenfunctions ranked by the alignment gap $A_i(\mathbf{y}\mathbf{y}^\top) - A_i(\mathbf{s}\mathbf{s}^\top)$: most left depicts eigenfunctions highly aligned with $\mathbf{s}$ while most right represents eigenfunctions highly aligned with $\mathbf{y}$. Particularly in the case of `Biased-MNIST` that the activation of $\mathcal{X}_s$ is prominently higher than that of $\mathcal{X}_y$ for eigenfunctions associated with spurious attribute. This might be due to the fact that in `Fashion-MNIST` $\mathcal{X}_y$ and $\mathcal{X}_s$ share common components, so certain eigenfunctions will rely on features from both domains.

## C.1 HYPERPARAMETERS

Figures C.6 and C.7 depict the performance across a parameter sweep, measured in terms of *average accuracy*, *worst-class accuracy*, and *worst-group accuracy*. The worst-class accuracy is defined as the accuracy of the poorest-performing class, calculated as $\min_{y' \in \mathcal{Y}} \mathbb{E}_{(x,y)|y=y'} \left[ \mathbb{1} \left[ f(x) = y \right] \right]$.

We observed a consistent trend where larger values of $\gamma$ and $0 < \beta < 1$ lead to the most significant enhancement in robustness. As recommended by Yang et al. (2023b), the results presented in Table 1 are based on *worst-class accuracy* on the validation set as the criteria.

| Selection | CMNIST | Fashion-MNIST | Biased-MNIST | Biased-CIFAR |
|---|---|---|---|---|
| average | $88.5_{\pm 1.3}$ | $76.8_{\pm 1.9}$ | $95.9_{\pm 0.9}$ | $75.9_{\pm 1.0}$ |
| worst-class | $88.5_{\pm 1.3}$ | $76.8_{\pm 1.9}$ | $95.9_{\pm 0.9}$ | $75.9_{\pm 1.0}$ |
| worst-group | $90.8_{\pm 1.0}$ | $78.5_{\pm 1.7}$ | $96.3_{\pm 0.7}$ | $79.6_{\pm 1.8}$ |
| oracle | $90.8_{\pm 1.0}$ | $78.8_{\pm 1.5}$ | $96.3_{\pm 0.7}$ | $79.6_{\pm 1.8}$ |

Table C.3: Test worst-group accuracy with different selection strategies on the validation set, where 'oracle' refers the best worst-group accuracy achieved on the test. In

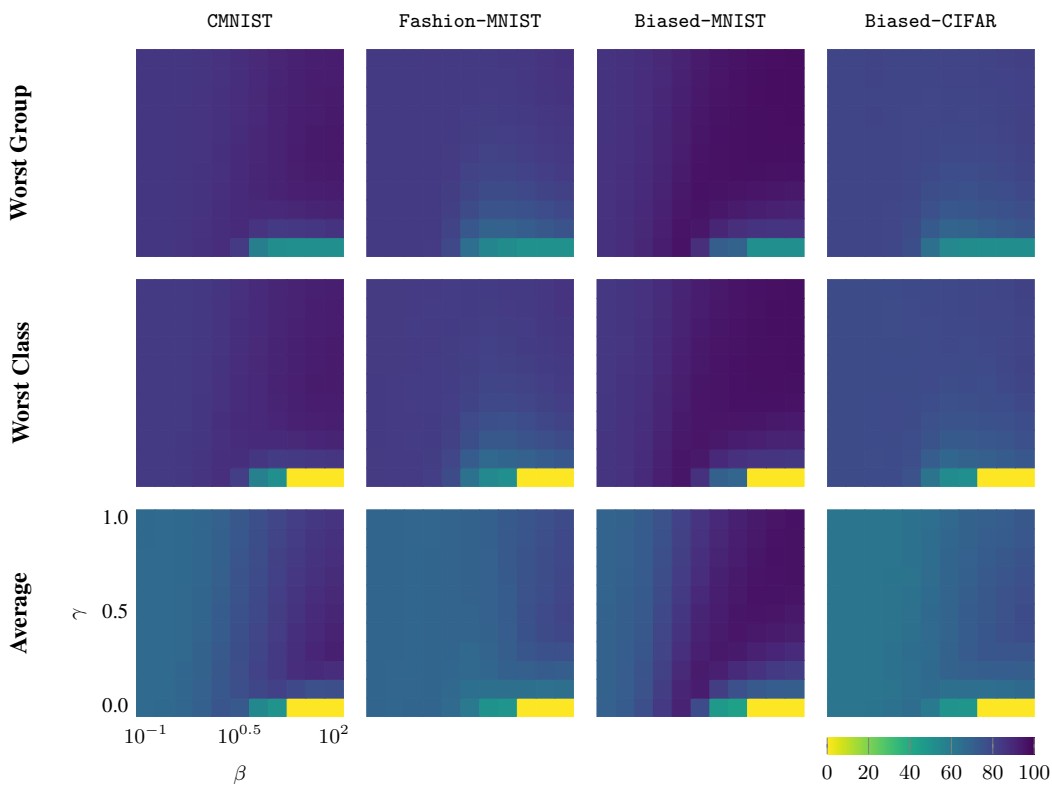

Figure C.6: Performance on the validation set with varying $\gamma$ ($x$-axis) and $\beta$ ($y$-axis).

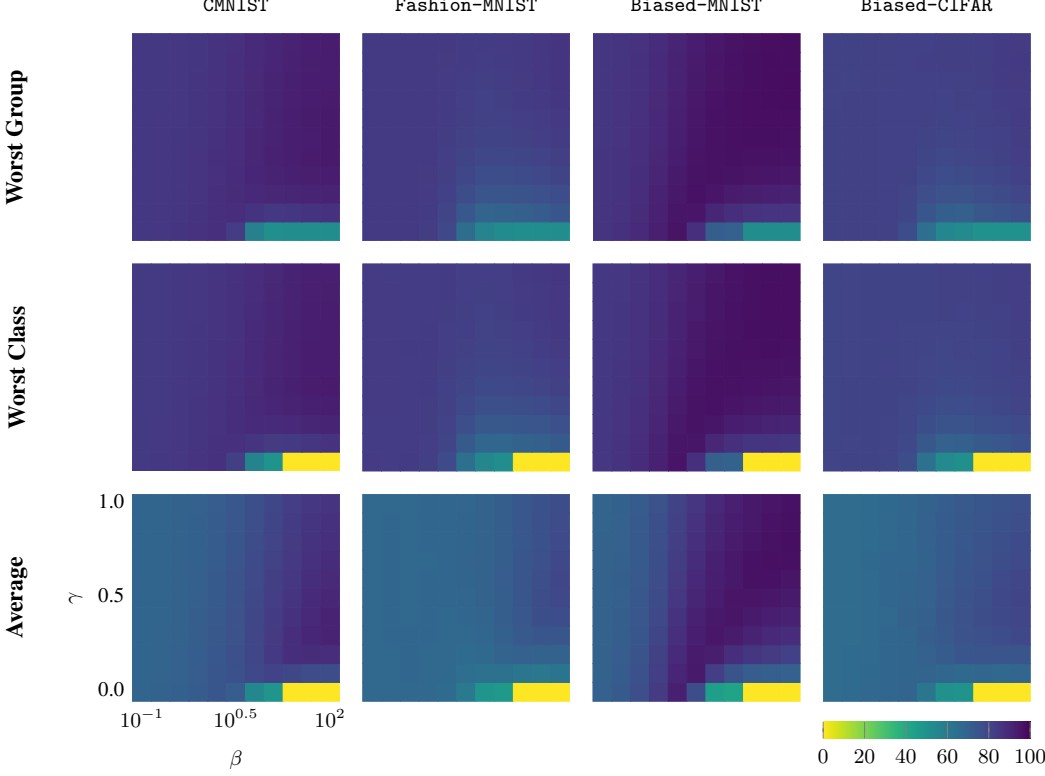

Figure C.7: Performance on the test set with varying $\gamma$ ($x$-axis) and $\beta$ ($y$-axis).

## C.2 DYNAMICAL SETTING

This section provides visualizations of the dynamics of training loss (Fig. C.8), training accuracy (Fig. C.9), test loss (Fig. C.10), and test accuracy (Fig. C.11) with an infinite-width network. In the presence of spurious correlations, all bias-conflicting subgroups tend to converge slowly on the loss compared to the bias-aligned subgroups, ultimately resulting in suboptimal performance on the test set.

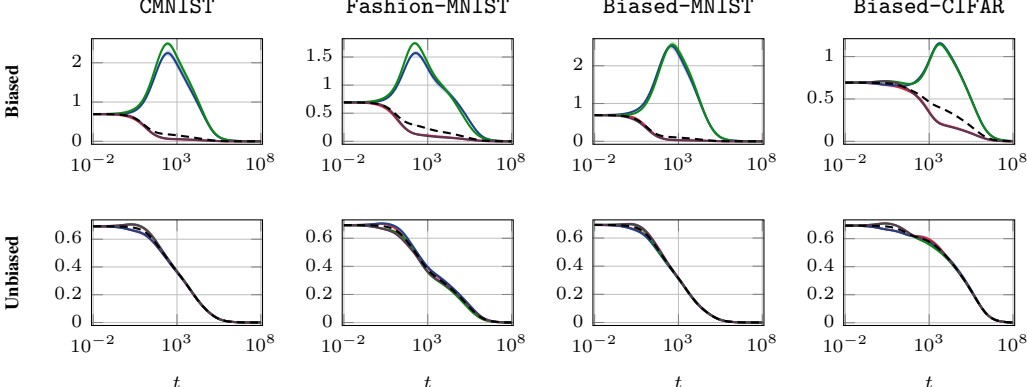

Figure C.8: training loss

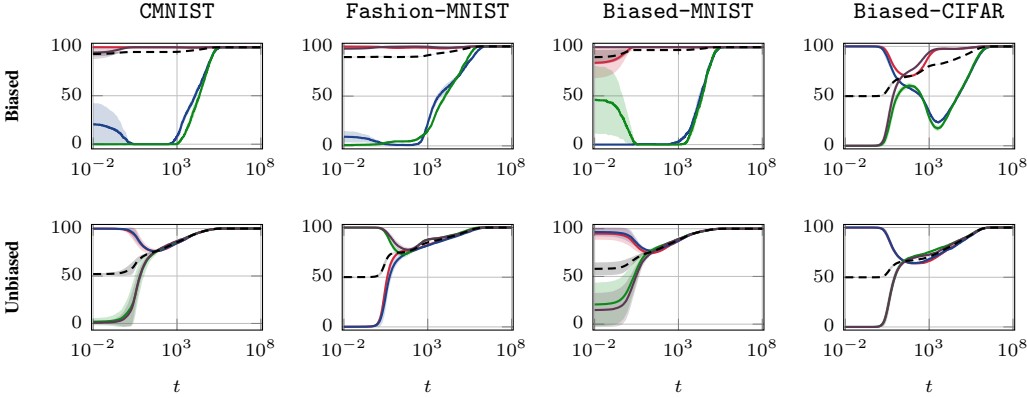

Figure C.9: training accuracy

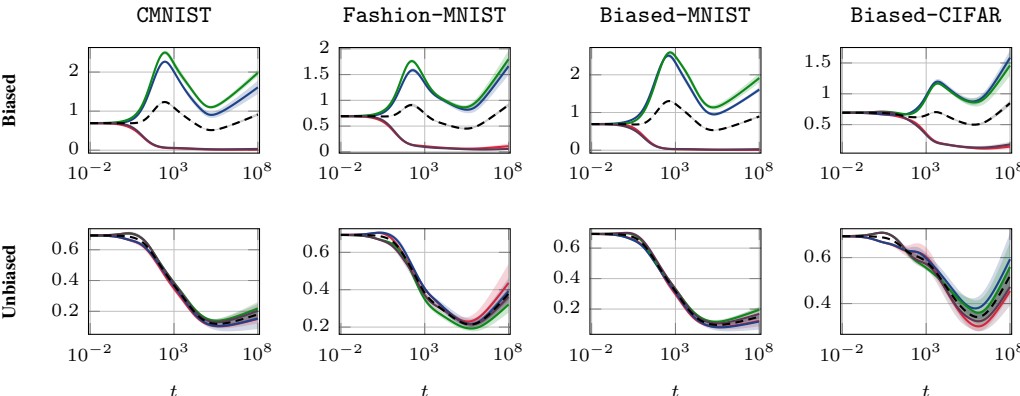

Figure C.10: test loss

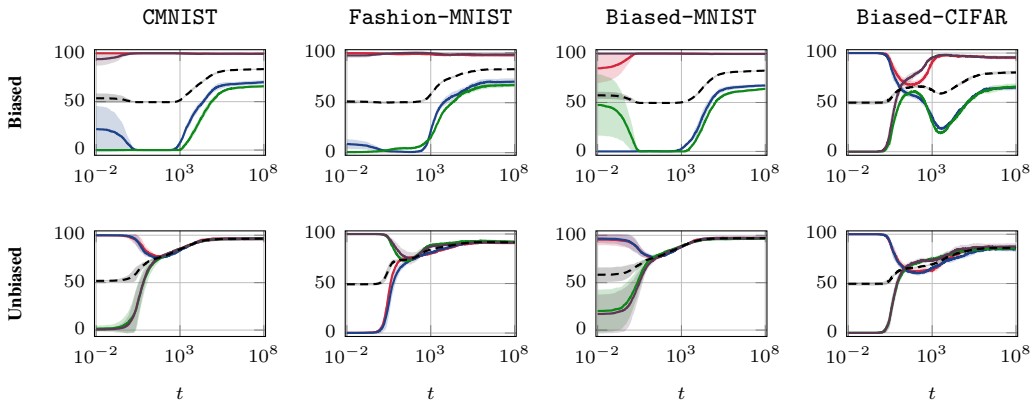

Figure C.11: test accuracy

## C.3 DYNAMICAL SETTING WITH MODIFIED SPECTRUM

This section provides the complete results of the dynamical setting (Eq. (7)) with the spectrum modification. Figures C.12 and C.13 present the performance on the training set, while Figs. C.14 and C.15 present the performance on the test set.

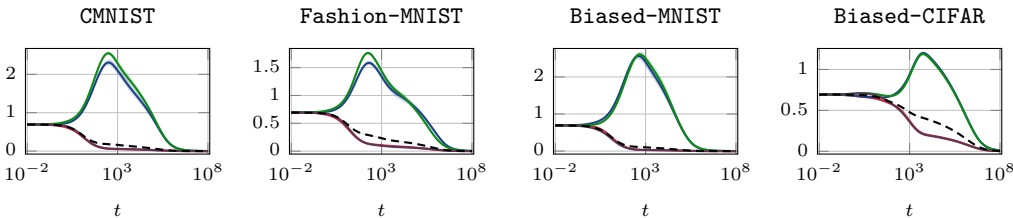

Figure C.12: training loss (with modified spectrum)

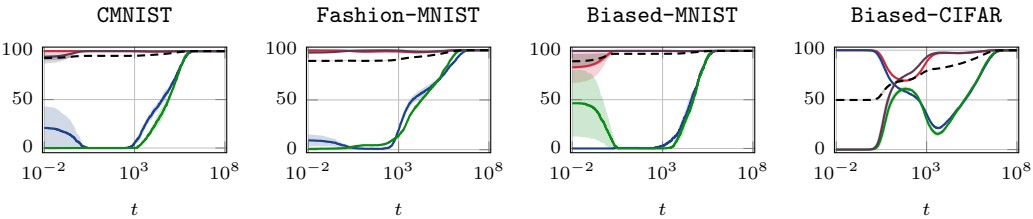

Figure C.13: training accuracy (with modified spectrum)

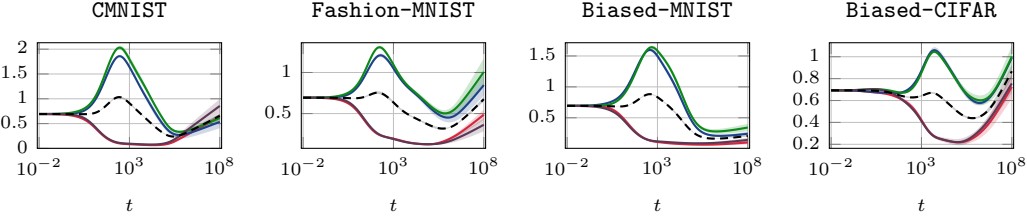

Figure C.14: test loss (with modified spectrum)

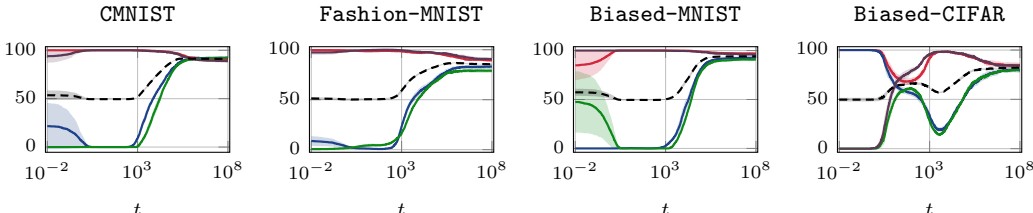

Figure C.15: test accuracy (with modified spectrum)

## C.4 FINITE WIDTH NEURAL NETWORKS

The learning curves depicted below align with the trends observed in Section C.2. Specifically, they illustrate that bias-aligned subgroups exhibit much faster convergence compared to the bias-conflicting subgroups and ultimately achieve performance levels above the average, while the latter tend to perform below the average. Table C.4 corresponds to Table C.2, in which we evaluate the performance of a finite-width DNN using its kernel representation. Figures C.16 to C.19 depict the corresponding results of Section C.2.

| Dataset | Target | Bias | Biased | | | Unbiased | | |
|---|---|---|---|---|---|---|---|---|
| | | | Avg. | Worst | $\Delta$ | Avg. | Worst | $\Delta$ |
| CMNIST | digit | color | $83.9_{\pm1.1}$ | $67.2_{\pm1.2}$ | $16.7_{\pm0.7}$ | $95.9_{\pm0.4}$ | $94.8_{\pm0.3}$ | $1.1_{\pm0.3}$ |
| | color | digit | $100.0_{\pm0.0}$ | $100.0_{\pm0.0}$ | $0.0_{\pm0.0}$ | $100.0_{\pm0.0}$ | $100.0_{\pm0.0}$ | $0.0_{\pm0.0}$ |
| Fashion-MNIST | fashion | digit | $76.0_{\pm1.6}$ | $51.2_{\pm2.7}$ | $24.7_{\pm1.8}$ | $84.3_{\pm0.6}$ | $81.5_{\pm1.5}$ | $2.7_{\pm1.0}$ |
| | digit | fashion | $100.0_{\pm0.0}$ | $99.9_{\pm0.1}$ | $0.0_{\pm0.1}$ | $99.9_{\pm0.1}$ | $99.8_{\pm0.2}$ | $0.1_{\pm0.1}$ |
| Biased-MNIST | digit | patch | $82.8_{\pm2.2}$ | $62.2_{\pm6.4}$ | $20.6_{\pm4.3}$ | $97.5_{\pm0.5}$ | $96.6_{\pm0.9}$ | $0.8_{\pm0.4}$ |
| | patch | digit | $99.6_{\pm0.5}$ | $98.5_{\pm2.0}$ | $1.1_{\pm1.5}$ | $99.9_{\pm0.2}$ | $99.7_{\pm0.3}$ | $0.2_{\pm0.2}$ |
| Biased-CIFAR | object | color patch | $72.5_{\pm2.6}$ | $59.1_{\pm1.7}$ | $13.4_{\pm2.5}$ | $74.6_{\pm2.6}$ | $71.7_{\pm1.5}$ | $2.9_{\pm1.8}$ |
| | color patch | object | $99.7_{\pm0.3}$ | $99.1_{\pm0.6}$ | $0.5_{\pm0.3}$ | $99.6_{\pm0.4}$ | $99.1_{\pm0.5}$ | $0.5_{\pm0.2}$ |

Table C.4: Performance with the empirical NTK.

| Dataset | Target | Bias | Biased | | | Unbiased | | |
|---|---|---|---|---|---|---|---|---|
| | | | Avg. | Worst | $\Delta$ | Avg. | Worst | $\Delta$ |
| CMNIST | digit | color | $86.7_{\pm1.3}$ | $72.2_{\pm2.1}$ | $14.5_{\pm1.1}$ | $97.6_{\pm0.3}$ | $96.7_{\pm0.3}$ | $0.9_{\pm0.4}$ |
| | color | digit | $100.0_{\pm0.0}$ | $100.0_{\pm0.0}$ | $0.0_{\pm0.0}$ | $100.0_{\pm0.0}$ | $100.0_{\pm0.0}$ | $0.0_{\pm0.0}$ |
| Fashion-MNIST | fashion | digit | $84.7_{\pm0.8}$ | $69.3_{\pm2.4}$ | $15.5_{\pm2.6}$ | $92.5_{\pm0.6}$ | $90.8_{\pm1.1}$ | $1.6_{\pm0.8}$ |
| | digit | fashion | $100.0_{\pm0.0}$ | $99.9_{\pm0.1}$ | $0.0_{\pm0.1}$ | $99.9_{\pm0.1}$ | $99.8_{\pm0.2}$ | $0.1_{\pm0.1}$ |
| Biased-MNIST | digit | patch | $87.5_{\pm1.6}$ | $73.5_{\pm4.0}$ | $14.0_{\pm2.5}$ | $98.0_{\pm0.3}$ | $97.5_{\pm0.5}$ | $0.6_{\pm0.4}$ |
| | patch | digit | $100.0_{\pm0.0}$ | $100.0_{\pm0.1}$ | $0.0_{\pm0.1}$ | $100.0_{\pm0.0}$ | $100.0_{\pm0.0}$ | $0.0_{\pm0.0}$ |
| Biased-CIFAR | object | color patch | $83.9_{\pm0.3}$ | $71.1_{\pm2.1}$ | $12.7_{\pm2.2}$ | $88.6_{\pm2.0}$ | $85.9_{\pm2.2}$ | $2.7_{\pm0.6}$ |
| | color patch | object | $100.0_{\pm0.0}$ | $100.0_{\pm0.0}$ | $0.0_{\pm0.0}$ | $100.0_{\pm0.0}$ | $100.0_{\pm0.0}$ | $0.0_{\pm0.0}$ |

Table C.5: Performance with SGD training.

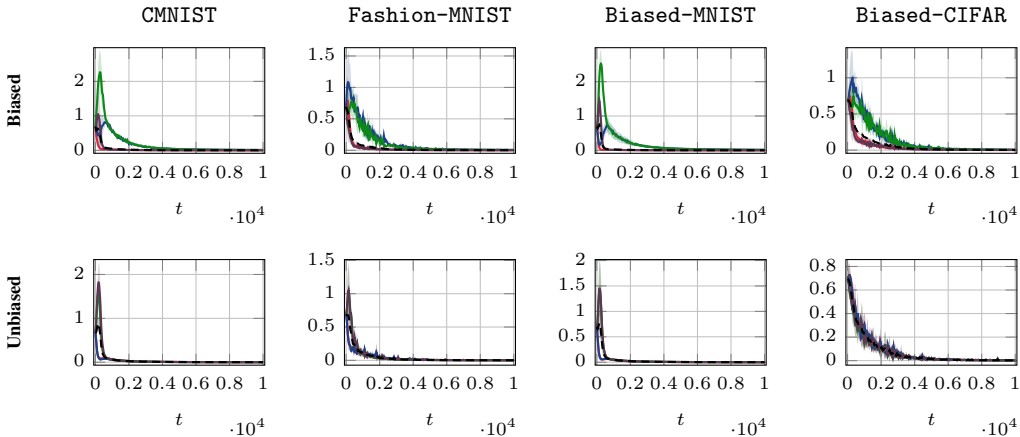

Figure C.16: training loss

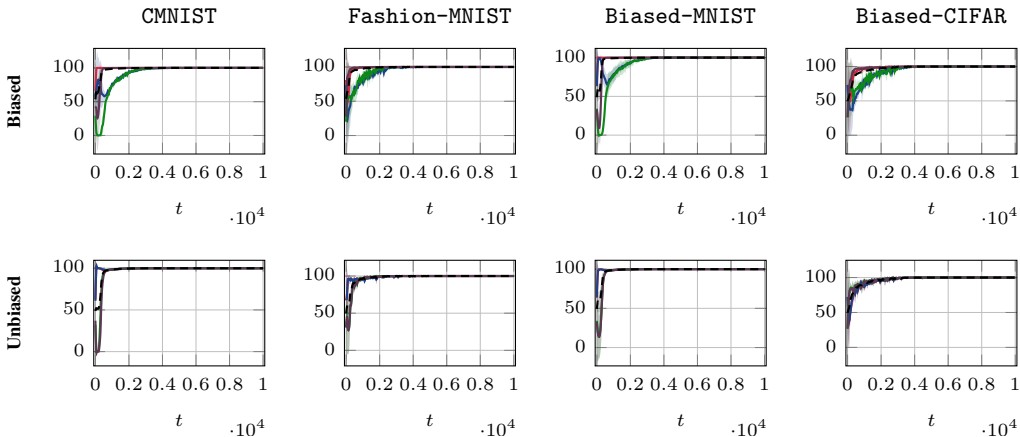

Figure C.17: training accuracy

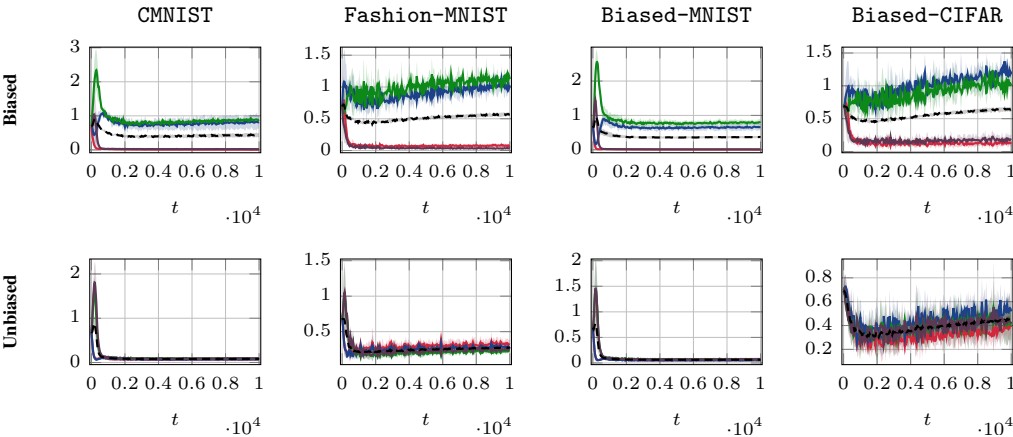

Figure C.18: test loss

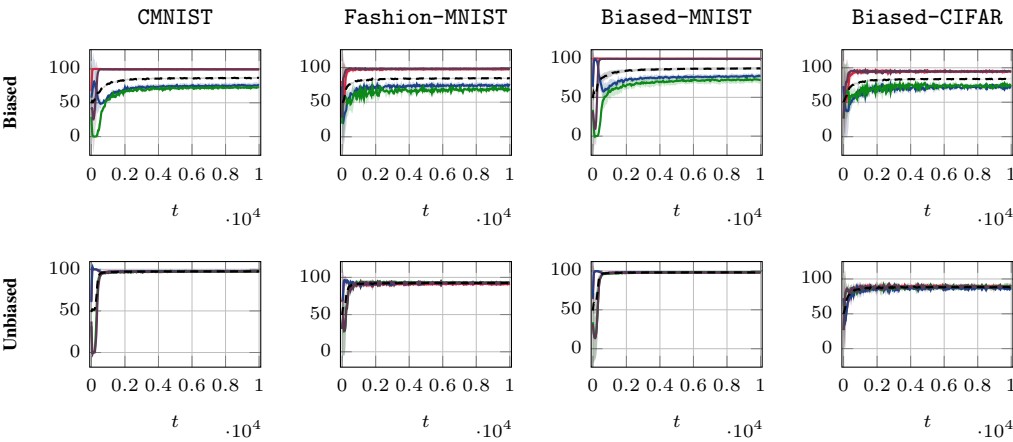

Figure C.19: test accuracy

