# OpenReview forum: "Understanding and addressing spurious correlation via Neural Tangent Kernels: A spectral bias perspective"
_ICLR.cc/2024/Conference — ICLR 2024 Conference Withdrawn Submission_

### Official Review · Reviewer_XMhp · 2023-10-27

**Soundness:** 3 good
**Presentation:** 4 excellent
**Contribution:** 3 good
**Rating:** 6
**Confidence:** 4

**Summary:**

The paper studies how spurious correlations are exploited and encoded in neural networks through the lens of the NTK in the infinite width limit, where the kernel governing the dynamics is fixed at initialization. This allows the training dynamics to be entirely predictable (in closed form), and allows for an eigendecomposition that allows for the extraction of eigenfunctions that are learned at different speeds based on the magnitude of the corresponding eigenvalue. In particular, through "saliency maps", the authors exploit this to identify that the leading eigenfunctions have high activations in the regions where the spurious features are present. This indicates that the network does make use of the spurious features to make predictions. The authors explain this in terms of the "simplicity bias" of the NTK: when the spurious feature can be explained with low-frequency functions, the signal will be picked up by the low-frequency eigenfunctions of the NTK, thus resulting in poor generalization. On the other hand, when the spurious feature is relatively complex compared to the generalizing signal, the model does generalize.

**Strengths:**

1. **Presentation**: The paper is very well-written, and the results are extremely well-presented. The logical argument of how spurious correlations affect generalization in the NTK regime flows very nicely from the author's considerations on previously observed phenomena such as simplicity bias, and adversarial robustness of the NTK.
2. **Quality**: the experiments are directly targeting the research question addressed by the authors. The visualization of the saliency map in Figure 2 explains well how the eigenfunctions encode the information of the spurious features. I also appreciated the in-depth discussion on the role of feature complexity, and how the spurious feature needs to be "simple" in order to cause problems in generalization for the NTK.
3. **Originality**. The problem of explaining spurious correlation under the NTK framework is novel, to my knowledge.

**Weaknesses:**

Main concern:
1. **Role of depth**. The authors state that deeper networks learn more complex features, and suggest that this can help to improve performances under the presence of spurious correlation as it is shown in Figure 5. However, it is unclear whether the performance improvement is due to the "feature complexity" argument put forth by the authors. There is no evidence provided that a similar improvement would happen even in the absence of the spurious feature. Also, there is no saliency map visualization confirming that the spurious features are not encoded in the leading eigenfunctions at larger depths. In Figure C.1, it is hard to confirm the authors' claim, as the gradient correlation seems to increase uniformly across all the groups.
Minor:
2. **Scope of the paper**. The paper adopts all existing techniques. For instance, saliency maps have been applied to NTK to study adversarial robustness (Tsilivis and Kempe, 2022), correctly cited by the paper. Also, the scope of the paper could have been more broadly enlarged to include the role of feature learning and changes in the NTK through training (some experiments are run in the Appendix, but the role of feature learning is not discussed). However, I consider these minor issues. I think that even without these minor improvements, the paper still deserves publication.

**Questions:**

1. Role of regularization. How does weight decay or other type of regularization affect the encoding of spurious features in the NTK?

---

> ### Author Response · Authors · 2023-11-17
> **Response to Reviewer XMhp**
>
> Thank you for your review. Your concerns will be addressed as follows:
>
> > There is no evidence provided that a similar improvement would happen even in the absence of the spurious feature
>
> As demonstrated by [1] in both theoretical and empirical contexts, elevating the depth (below the optimal depth) increases the capacity to learn complex features. This is attributed to the condition number approaching one with increasing depth, signifying a faster convergence rate for high-frequency components. Importantly, this argument remains valid irrespective of the existence of spurious correlations.
>
> > In Figure C.1, it is hard to confirm the authors’ claim, as the gradient correlation seems to increase uniformly across all the groups
>
> As depicted in Figure 6, once surpassing the optimal depth, the robustness diminishes. Figure C.1 explains this phenomenon, revealing that deeper models result in a broader spectrum that inevitably captures noise. Consequently, gradient correlations become similar across all groups.
>
> > Role of regularization. How does weight decay or other type of regularization affect the encoding of spurious features in the NTK?
>
> Thank you for the suggestion. In this work, we have not considered the effect of regularization. However, it has been thoroughly studied by [2], and their work will serve as a valuable reference for our future work.
>
> ---
> [1] G. Yang et al. A Fine-Grained Spectral Perspective on Neural Networks https://arxiv.org/abs/1907.10599
>
> [2] Jaehoon Lee et al. Finite Versus Infinite Neural Networks: An Empirical Study http://arxiv.org/abs/2007.15801

---

> > ### Comment · Reviewer_XMhp · 2023-11-21
> > **Answer to rebuttal**
> >
> > I thank the authors for their explanation regarding the role of depth. I would still suggest performing a preliminary experiment where the saliency map is plotted at multiple points during training to see the effect of feature learning. This would give more food for thought and motivate future work. I am keeping my score for now.

---

### Official Review · Reviewer_hgH8 · 2023-10-31

**Soundness:** 1 poor
**Presentation:** 3 good
**Contribution:** 1 poor
**Rating:** 3
**Confidence:** 4

**Summary:**

The paper focuses on the problems posed by spurious correlations when learning with the neural tangent kernel and proposes a technique for improving NTK robustness when such correlations exist. The paper introduces a model of spurious correlation, where true labels $Y$ are independent of spurious features $S$ at test time, but have a planted dependence at training time modulated by parameter $\alpha$. They consider several tasks that combine spurious features from a simpler learning task with real features from a more difficult task.

They demonstrate empirically that the spurious features are most strongly associated with low-frequency eigenvectors of the NTK matrix $H$ by giving a visualization of salience maps for each feature and computing eigenvector alignment scores. Given the association between spurious features and low-frequency features, they propose a modification of the kernel that scales eigenvalues to reduce the impact of low-frequency features. They demonstrate empirically that this approach improves generalization performance on the tasks and erases the gap between performance on different subgroups (characterized by the label $Y$ and the spurious features $S$). The appendix includes a hyper-parameter sweep over the kernel modification parameters $\gamma$ and $\beta$.

**Strengths:**

The paper is overall well-written and its survey of the robustness literature appears to be strong. The NTK background is presented in an intuitive and understandable way. The tasks are cleanly presented and dataset visualizations in Figures 1 and 3 are useful. The thoroughness demonstrated by the hyper-parameter sweeps in Figures C.6 and C.7 is appreciated. The problem is certainly compelling, and the idea of tuning the spectrum of the kernel matrix is interesting. While I find the connections to deep neural networks limited, I think this work is a good foundation for a rigorous experimental study of spurious bias reduction for kernel methods.

**Weaknesses:**

I have two high-level critiques of the work, which I would be interested to hear the authors respond to. The first pertains to the relevance of the NTK and the methods of this work to neural networks, and the second to the generality of the paper's claims about the spectrum of spurious features.

## Relevance of NTK and algorithm to DNNs

The authors motivate their work by introducing the important issue of spurious correlations in deep neural networks, but they restrict their focus to the NTK approximation of a neural network. While the neural tangent kernel is an appealing model that allows one to apply convex analysis tools to deep learning, both empirical and theoretical work shows that the NTK fails to match neural network performance and cannot perform key deep learning functions like feature learning; consider reading and citing [works](https://papers.nips.cc/paper_files/paper/2019/hash/c133fb1bb634af68c5088f3438848bfd-Abstract.html) [like](https://arxiv.org/abs/2011.14522) [these](https://arxiv.org/abs/2206.10012). The NTK and kernel methods broadly are certainly worthy of study, but this paper does not address the critiques of these models in their introduction or NTK discussion.

The feature learning critique of the NTK is a particular issue for this paper because the methods use the spectrum of the kernel matrix at initialization, without accounting for the fact that gradient descent tends to encode task-dependent features in the bottom layers of neural networks. Thus, the features analyzed in this work are unlikely to correspond to the actual features encoded in a trained neural network. In addition, the mitigation algorithm presented in the paper only applies when the optimization algorithm has direct access to the kernel, and it doesn't appear to work for neural networks trained with gradient descent.

Perhaps the paper could be more strongly motivated if it were grounded in studying spurious correlations in kernel methods without needing the analogy to DNNs to hold? If the paper is to focus on DNNs, it needs to include a stronger accounting of the limitations of the NTK and a proposal for how it can be applied the feature learning that occurs in neural networks trained with gradient descent.

## Generality of low-frequency spurious features

I am concerned that the principal claim of the paper---that spurious correlations are aligned with low-frequency eigenvectors of the kernel matrix and that they can be mitigated by down-weighting the respective eigenvalues---is an artifact of the datasets considered in Sections 4 and 5. Specifically, the five datasets considered have artificially planted spurious features (e.g. digit color) that are far easier to learn than the true labels (e.g. 0-4 vs 5-9). The dominance of the spurious labels in the low-frequency features of the biased data is made evident for CMNIST and Biased-MNIST in Figure 4, but it's less strong for other tasks, and it's unclear whether this holds for "real world" datasets with spurious features.

As a result, it's unclear whether the algorithmic approach in Section 5 would perform well outside the collection of tasks considered. The paper does little to rule out the possibility that flattening the spectrum of $\Lambda$ with $\nu$ will cause problems on other tasks, since it's conceivable that salient features could be lower frequency elsewhere. For this paper to be more compelling, I would like to see whether these results still hold on less artificial datasets, such as Waterbirds and CelebA, which were used in addition to the MNIST-type tasks to evaluate the methods of [the](https://arxiv.org/abs/2210.00055) [papers](https://openreview.net/pdf?id=Zb6c8A-Fghk) they take dataset inspiration from.

## Other significant concerns

Several figures in the paper have significant presentation issues.
- Figures 4 and 5 are missing numerical axes.
- Using the normalized index $k$ as the x-axis in Figures 3 and 5 adds some confusion, since it's unclear what number of features is represented by $k$, as it's given as a fraction.
- There's no discussion in Appendix B about how the choices of planted bias $\alpha$ were chosen and why choices differ for each dataset.
- The CIFAR-MNIST task is introduced in Section 4.1 without being referenced elsewhere in the paper body.

## Minor comments
- In Section 3.1, the actual datasets do not strictly belong to the regime where $\mathcal{X} = \mathcal{X}_y \times \mathcal{X}_s$, since $\mathcal{X}$ defines the input space, and several tasks (e.g. CMNIST) have both relevant and spurious features operating on the same pixels.
- There is some confusing mathematical notation in Section 3.2. $\dot{\theta}_t$ is somewhat atypical notation, since $\theta_t$ is the typical presentation of a step of a discrete dynamical system, rather than $\theta(t)$ for continuous systems; perhaps $\dot{\theta}(t)$ is more appropriate? Equation (3) defines the time derivative as a discrete-time gradient step, which is slightly confusing without limit notation.
- In Section 4.1, the terms "bias-aligned" and "bias-conflicting" are introduced, but are only used in the appendix.
- In Section 4.2 in the sentence starting with "Based on the decomposition in Eq (10), training a DNN with gradient descent...," I would recommend adding the qualification "in the lazy or NTK regime."
- The caption of Table C.3 ends with an incomplete sentence.

**Questions:**

Can the approaches presented in the paper by applied to all kernel methods, rather than just the NTK? Given the aforementioned issues of the analogy between the NTK and neural networks, perhaps the paper could be motivated more strongly as a method for kernel methods, rather than neural networks.

Is the model of spurious correlation introduced in Section 3.1 novel? If not, I would recommend adding citations to where it's defined in the literature, and if so, I would recommend explaining how it differs from other formulations.

Did you consider including theoretical results about the effectiveness of the algorithm? I suspect that there's a natural theorem to prove about when the approach will provably work for some reasonable data distribution assumptions on the spurious features aligning with the low-frequency eigenvectors.

---

> ### Author Response · Authors · 2023-11-17
> **Response to Reviewer hgH8**
>
> Thank you for your review. Concerns related to presentation and minor comments will be resolved in the updated version.
>
> > Relevance of NTK and algorithm to DNNs
>
> We agree that the NTK has its limitations and may not fully match the performance of neural networks in certain aspects. We will revise our introduction and NTK discussion to better acknowledge and address these limitations. Despite the absence of feature learning, NTKs have been extensively utilized [1-5] for understanding and improving model performance. Importantly, the additional results (refer to our response to Reviewer cfRQ) highlight the effectiveness of our method for the learned features, analogous to fine-tuning [1].
>
> > Generality of low-frequency spurious features
>
> Please refer to our response to Reviewer cfRQ for additional results on Waterbirds and CelebA datasets.
>
> > Is the model of spurious correlation introduced in Section 3.1 novel? If not, I would recommend adding citations to where it's defined in the literature, and if so, I would recommend explaining how it differs from other formulations.
>
> The model mentioned in Section 3.1 is a well-established notion of spurious correlation [6, 7]. We appreciate the reviewer for bringing this to our attention, and we will include some relevant references in the revised version.
>
> ---
>
> [1] Sadhika Malladi et al. A Kernel-Based View of Language Model Fine-Tuning https://arxiv.org/abs/2210.05643
>
> [2] Chia-Hung Yuan and Shan-Hung Wu. Neural Tangent Generalization Attacks https://proceedings.mlr.press/v139/yuan21b.html
>
> [3] Nikolaos Tsilivis and Julia Kempe. What Can the Neural Tangent Kernel Tell Us About Adversarial Robustness? http://arxiv.org/abs/2210.05577
>
> [4] Haonan Wang et al. Deep Active Learning by Leveraging Training Dynamics https://arxiv.org/abs/2110.08611
>
> [5] Mohamad Amin Mohamadi, Wonho Bae, and Danica J. Sutherland. Making LookAhead Active Learning Strategies Feasible with Neural Tangent Kernels https://arxiv.org/abs/2206.12569
>
> [6] Robert Geirhos et al. Shortcut Learning in Deep Neural Networks https://arxiv.org/abs/2004.07780
>
> [7] Yuzhe Yang et al. Change Is Hard: A Closer Look at Subpopulation Shift https://arxiv.org/abs/2302.12254

---

> > ### Comment · Reviewer_hgH8 · 2023-11-23
> >
> > I thank the authors for their response and for their willingness to run additional experiments.
> >
> > While I agree with the authors that the NTK remains generally relevant to some extent and that there are some circumstances where training dynamics without feature learning are worth studying, I maintain my belief that feature learning is a particularly central dynamic to consider when evaluating spurious features in neural nets.
> >
> > I appreciate the work done to implement the additional experiments, but the sharp reduction in average accuracy for Waterbirds and the almost negligible impact on CelebA do not change my current views about the generality of low-frequency features.
> >
> > As of now, I maintain my score.

---

### Official Review · Reviewer_cfRQ · 2023-11-01

**Soundness:** 2 fair
**Presentation:** 3 good
**Contribution:** 1 poor
**Rating:** 3
**Confidence:** 4

**Summary:**

This paper establishes a connection between spectral bias of neural networks and how that effects their subgroup robustness. They first analyze the eigenfunctions of the NTK Gram matrix and show that features that are spurious correspond to lower-order eigenfunctions, which correspond to larger eigenvalues. When such features are entangled with the label, it can cause the model to rely on these more strongly because eigenfunctions with larger eigenvalues are learned faster. In order to mitigate this effect, the authors propose a modification to the Gram matrix, to make the eigenvalues more similar to each other and encourage learning diverse features. They use some simple variants of MNIST and CIFAR datasets to demonstrate their observations and evaluate their approach.

**Strengths:**

1. The paper presents a connection between spectral bias of NNs and their tendency to learn spurious features which leads to low subgroup robustness, which is interesting.

2. They propose an approach to improve subgroup robustness that seems effective on some variants of MNIST and CIFAR.

3. The paper is well-written and easy to follow.

**Weaknesses:**

1. ****Although the connection between spectral bias and subgroup robustness is nice, it doesn't really offer much new insights.****

From the spectral bias perspective, there are several works [2,3], which have theoretically shown that eigenvectors of the Gram matrix with larger eigenvalues correspond to 'simpler' features, using some notions of simplicity. From the subgroup robustness perspective, there are works [4,5] that show that spurious features are simpler and simplicity bias of NNs causes them to rely more on such features. Based on these works, one can expect that spurious features would correspond to lower order eigenfunctions which will have higher eigenvalues.

2. ****The idea to make the eigenvalues of the NTK Gram matrix more balanced to improve subgroup robustness is not new (see [1]). The proposed approach is not scalable.****

The paper is missing discussion about a very closely related work [1]. [1] mainly talks about how spectral bias can cause NNs to rely strongly on a subset of features, and also proposes an approach for spectral decoupling (making the eigenvalues of the NTK Gram matrix more balanced). They show that regularizing the logits of the model can help do that (in some simple settings) and also empirically validate their approach on a range of datasets (including subgroup robustness datasets). Their approach is also more efficient and scalable than the approach presented in the current work, which does not scale to large datasets. Based on this, the proposed method doesn't seem very valuable.

3. ****Limited evaluation.****

The paper considers MNIST and CIFAR datasets, but evaluation on the usual subrgoup robustness benchmarks (CelebA and Waterbirds) seems missing. The paper also does not compare their approach with any other method to improve subgroup robustness.


****References:****

[1] M. Pezeshki et al. ****Gradient Starvation: A Learning Proclivity in Neural Networks**** https://arxiv.org/abs/2011.09468

[2] Y. Cao et al. ****Towards Understanding the Spectral Bias of Deep Learning**** https://arxiv.org/abs/1912.01198

[3] G. Yang et al. ****A Fine-Grained Spectral Perspective on Neural Networks**** https://arxiv.org/abs/1907.10599

[4] Y. Yang et al. ****Identifying Spurious Biases Early in Training through the Lens of Simplicity Bias**** https://arxiv.org/abs/2305.18761

[5] H. Shah et al. ****The Pitfalls of Simplicity Bias in Neural Networks**** https://arxiv.org/abs/2006.07710

**Questions:**

(See weaknesses above)

Can the authors discuss the contributions of their work in light of this related work [1-5]? I suggest doing a more thorough review of related work and including a detailed discussion on the contributions and what are the insights that are new compared to these other works. And particularly, what is the contribution of this work compared to [1]?

---

> ### Author Response · Authors · 2023-11-17
> **Response to Reviewer cfRQ**
>
> Thank you for your review. Your concerns will be addressed as follows:
>
> > Although the connection between spectral bias and subgroup robustness is nice, it doesn’t really offer much new insights.
>
> Building upon [2,3], our work aims to tackle the issue of spurious correlation. Particularly in our context, the class labels are entangled with simpler features, leading to challenges in the model's ability to learn semantic representations. We agree that [4], [5] and our work are based on the similar concept, i.e., spurious features induce a simplistic bias that degrades the performance of the model. However, none of these studies formulate their investigation within the NTK regime, nor do they present explicit evidence supporting Reviewer’s statement ”one can expect that spurious features would correspond to lower-order eigenfunctions which will have higher eigenvalues.” In addition to that, [4] proposed the reweighing training set approach, a common technique to address spurious correlation problems, as discussed in the related works section. Our work offers a novel perspective, suggesting that robustness can be achieved without altering the training distribution or the training objective. Instead, it emphasizes that modifying the model’s properties can achieve this objective
>
> > The idea to make the eigenvalues of the NTK Gram matrix more balanced to improve subgroup robustness is not new (see [1]). The proposed approach is not scalable.
>
> Our work attributes the impact of spurious correlation to spectral bias. While spectral decoupling [1] is theoretically scalable, we observed that practical implementation requires distinct treatments for the logits of each class, (requiring hyperparameters per class) and thus elevating the implementation complexity.
>
>
> > Limited evaluation.
>
> The following are the results for the Waterbirds and CelebA datasets, and our approach achieves the state-of-the-art performance (cf. Appendix E.3 in [7]). The experiment follows a linear probing approach (similar to [6]), using feature embeddings extracted from a trained model as inputs to the NTK. It's worth noting that the consideration of linear probing as a solution has also been discussed in recent NTK literature [8].
>
> **Waterbids**
> |                   |   Avg. Acc   |  Worst Acc.  |   $\Delta$   |
> |:-----------------:|:------------:|:------------:|:------------:|
> |     original      | $92.4\pm0.3$ | $72.9\pm0.8$ | $19.5\pm0.8$ |
> | modified spectrum | $86.1\pm1.2$ | $80.0\pm3.0$ | $6.1\pm2.3$  |
>
> **CelebA**
> |                   |   Avg. Acc   |  Worst Acc.  |   $\Delta$   |
> |:-----------------:|:------------:|:------------:|:------------:|
> |     original      | $99.0\pm0.1$ | $88.9\pm3.0$ | $10.1\pm3.0$ |
> | modified spectrum | $99.0\pm0.1$ | $89.0\pm3.2$ | $9.9\pm3.1$  |
>
> Note that, for the CelebA dataset, we used a subset of the training split and the entire evaluation split, following the same procedure as outlined in [1] (refer to Appendix B.5.1 in [1]).
> We observed superior performance with the orignal NTK. Importantly, modifying the spectrum does not lead to a deterioration in performance.
>
>
> > Can the authors discuss the contributions of their work in light of this related work [1-5]?
>
> We appreciate Reviewer for highlighting the work of [1], and we are well aware of this study. We will discuss it in our related work section. References [2-5] are cited in the related work section, and we will provide more detailed elaboration on the connection to these prior works.
>
> ---
>
> [1] M. Pezeshki et al. Gradient Starvation: A Learning Proclivity in Neural Networks https://arxiv.org/abs/2011.09468
>
> [2] Y. Cao et al. Towards Understanding the Spectral Bias of Deep Learning https://arxiv.org/abs/1912.01198
>
> [3] G. Yang et al. A Fine-Grained Spectral Perspective on Neural Networks https://arxiv.org/abs/1907.10599
>
> [4] Y. Yang et al. Identifying Spurious Biases Early in Training through the Lens of Simplicity Bias https://arxiv.org/abs/2305.18761
>
> [5] H. Shah et al. The Pitfalls of Simplicity Bias in Neural Networks https://arxiv.org/abs/2006.07710
>
> [6] Pavel Izmailov et al. On Feature Learning in the Presence of Spurious Correlations https://arxiv.org/abs/2210.11369
>
> [7] Yuzhe Yang et al. Change Is Hard: A Closer Look at Subpopulation Shift https://arxiv.org/abs/2302.12254
>
> [8] Sadhika Malladi et al. A Kernel-Based View of Language Model Fine-Tuning https://arxiv.org/abs/2210.05643

---

> > ### Comment · Reviewer_cfRQ · 2023-11-23
> >
> > I thank the authors for providing a detailed response and for sharing additional experimental results. I still have the following concerns and as of now, I will maintain my score.
> >
> > 1. While prior work does not present explicit evidence that spurious features correspond to lower order eigenvectors with larger eigenvalues, this work does not really offer new insights. Based on the findings in prior work [1-5], one can expect such a behaviour. It is important to discuss the insights and contribution of this work compared to [1-5].
> >
> > 2. The authors of [1] use a single hyperparameter for the logit regularization and not specific values for each class. The statement about the increase in implementation complexity due to additional hyperparameters does not seem correct.
> >
> > 3. Thank you for sharing additional results on Waterbirds and CelebA. The statement that these results are state of the art is not correct, there are methods in App. E.3 in [7] (cited by the authors) that are better in terms of worst-group accuracy and the gap between average and worst-group accuracy. Also, the improvement in CelebA is negligible. This does not address the concern about the usefulness of the approach (given that the approach is also not scalable).